# Prostate-Specific Membrane Antigen-Targeted Therapy in Prostate Cancer: History, Combination Therapies, Trials, and Future Perspective

**DOI:** 10.3390/cancers16091643

**Published:** 2024-04-25

**Authors:** Francesco Mattana, Lorenzo Muraglia, Antonio Barone, Marzia Colandrea, Yasmina Saker Diffalah, Silvia Provera, Alfio Severino Cascio, Emanuela Omodeo Salè, Francesco Ceci

**Affiliations:** 1Division of Nuclear Medicine, IEO European Institute of Oncology IRCCS, 20141 Milan, Italy; antonio.barone@ieo.it (A.B.); marzia.colandrea@ieo.it (M.C.); alfio.cascio@ieo.it (A.S.C.); francesco.ceci@ieo.it (F.C.); 2Division of Nuclear Medicine, Humanitas IRCCS, 20141 Milan, Italy; lorenzo.muraglia@cancercenter.humanitas.it; 3Division of Nuclear Medicine, Hospital Clínico Universitario Lozano Blesa, 50009 Zaragoza, Spain; yasminasaker.d@gmail.com; 4Division of Pharmacy, IEO European Institute of Oncology IRCCS, 20141 Milan, Italy; silvia.provera@ieo.it (S.P.); eomodeo@ieo.it (E.O.S.); 5Department of Oncology and Hemato-Oncology, University of Milan, 20122 Milan, Italy

**Keywords:** PSMA, prostate cancer, lutetium, targeted therapy, radioligand therapy, oncology

## Abstract

**Simple Summary:**

Radioligand therapy plays a crucial role in the management of prostate cancer patients, for whom despite all available treatments, natural progression is almost inevitable. The failure of therapeutic options is likely to refer to the intrinsic tumor heterogeneity and the development of oncologic resistance pathways. To address this resistance, different trials are attempting to study the effectiveness and safety of combined therapies. This review provides a comprehensive overview of the current and future applications of radioligand therapy in prostate cancer from its initial application, moving towards future perspectives, and encompassing the main characteristics of ongoing trials related to this topic.

**Abstract:**

In the last decades, the development of PET/CT radiopharmaceuticals, targeting the Prostate-Specific Membrane Antigen (PSMA), changed the management of prostate cancer (PCa) patients thanks to its higher diagnostic accuracy in comparison with conventional imaging both in staging and in recurrence. Alongside molecular imaging, PSMA was studied as a therapeutic agent targeted with various isotopes. In 2021, results from the VISION trial led to the Food and Drug Administration (FDA) approval of [^177^Lu]Lu-PSMA-617 as a novel therapy for metastatic castration-resistant prostate cancer (mCRPC) and set the basis for a radical change in the future perspectives of PCa treatment and the history of Nuclear Medicine. Despite these promising results, primary resistance in patients treated with single-agent [^177^Lu]Lu-PSMA-617 remains a real issue. Emerging trials are investigating the use of [^177^Lu]Lu-PSMA-617 in combination with other PCa therapies in order to cover the multiple oncologic resistance pathways and to overcome tumor heterogeneity. In this review, our aim is to retrace the history of PSMA-targeted therapy from the first preclinical studies to its future applications in PCa.

## 1. Introduction

Prostate cancer (PCa) is one of the most frequently diagnosed cancers in men and the fifth leading cause of cancer death for males annually [1]. Even if a curative intent is feasible in patients with localized or locally advanced PCa, most patients experience a recurrence. The natural history of PCa consists of the evolution from a localized to a metastatic setting, but several possible life-prolonging therapies are available [1,2]. For metastatic hormone-naïve prostate cancer (mHNPC), androgen deprivation therapy (ADT) (e.g., leuprolide, goserelin, triptorelin, degarelix, and relugolix) can be used to decrease testosterone levels and, consequently, inhibit one of the most important growth pathways of PCa cells [3]. More recently, in this clinical setting, some studies showed that the combination of different therapies can prolong patients’ survival. ADT has been combined with chemotherapy [4,5], with a novel androgen receptor pathway inhibitor (ARPI) (e.g., abiraterone, enzalutamide, or apalutamide) [6,7,8], or even used inside a triplet regimen treatment along with the concomitant administration of ARPI (abiraterone or darolutamide) plus docetaxel [9,10].

According to EAU guidelines, in metastatic castration-resistant prostate cancer (mCRPC), the recommended treatments are taxane-based drugs (docetaxel and cabazitaxel), ARPI, [^223^Ra]Ra-dichloride, immunotherapy, radioligand therapies (RLTs), and olaparib (in patients with BRCA1/BRCA2 mutations) [1]. In mCRPC, olaparib and rucaparib are licensed by the FDA (European Medicines Agency only approved olaparib), while several other PARP inhibitors are under investigation (e.g., talazoparib, niraparib) [1]. In 2004, docetaxel demonstrated to prolong survival in two phase III trials [11,12], while abiraterone and enzalutamide are life-prolonging treatments in mCRPC patients when used both as first-line therapy and post-docetaxel [13,14]. Cabazitaxel is FDA-approved in mCRPC patients after progression on docetaxel [15] and demonstrated to improve various clinical outcomes in patients previously treated with docetaxel and ARPI (abiraterone or enzalutamide) compared with retreatment with the alternative ARPI [16]. In a phase III trial, [^223^Ra]Ra-dichloride, an α emitter that selectively targets bone metastases, improved overall survival (OS) compared with placebo, in addition to the best standard of care, in men with castration-resistant PCa and bone metastases [17]. In a multicenter phase III trial, Sipuleucel-T, an autologous active cellular immunotherapy, prolonged OS among asymptomatic or minimally symptomatic mCRPC patients (predominantly before chemotherapy) compared with placebo [18]. In mCRPC patients with at least one alteration in BRCA1, BRCA2, or ATM genes, olaparib prolongs OS compared with enzalutamide or abiraterone plus prednisone as control therapy [19]. Unfortunately, there is still a need to find new and more effective therapeutic options because of the high percentage of PCa progressions despite all these available treatments.

In the latest decades, the development of PET/CT radiopharmaceuticals, targeting the Prostate-Specific Membrane Antigen (PSMA), changed the management of PCa patients thanks to its higher diagnostic accuracy compared with conventional imaging both in staging and recurrence [20]. Thanks to the overexpression of PSMA in most histotypes of PCa cells, in addition to the low expression in normal tissue, PSMA became a crucial target also for therapeutic purposes [21].

In this review, we retrace the history of PSMA-target therapy and the role and and the main characteristics of the trials that led to Food and Drug Administration (FDA) approval of [^177^Lu]Lu-PSMA-617 as a therapeutic agent for metastatic PCa. Moreover, we summarize the main combination trials of [^177^Lu]Lu-PSMA-617 with other agents that will lead the way for future directions of PCa medical treatment.

## 2. PSMA-Targeted Therapies: From the Origins to Our Days

### 2.1. The Origins: Preclinical Studies

Research into newer diagnostics and treatments for PCa surged in 1993 when the Prostate Cancer Foundation (PCF) started funding projects aimed at a deeper understanding of a recently discovered transmembrane protein highly expressed in the prostate gland: PSMA.

This story takes us from the streets of Santa Monica, Los Angeles, California, where the headquarters of the PCF is located, to the streets of Lenox Hill, Manhattan, New York, where the first antibodies against the extracellular domain of PSMA were developed at Weill Cornell Medical College in 1997 [22]. One year later, the same group discovered that the PSMA receptor was endocytosed from the surface inside the cell and, therefore, might be used to detect PCa cells. The rate of PSMA internalization was increased by the use of anti-PSMA antibodies [23]. These results established the scientific basis for the development of a new targeted therapy.

In 2000, at Memorial Sloan Kettering Cancer Center, Sheinberg et al. [24] studied the in vitro and in vivo use of a novel α-particle emitting monoclonal antibody (J591) targeting the external domain of PSMA, which showed a significant prostatic-specific antigen (PSA) reduction and higher tumor-free survival in mice. After this first experience, a series of phase I and II clinical trials testing the use of the humanized version of J591 was initiated at Weill Cornell. These studies also represented the first use of lutetium-177 (^177^Lu) as a systemically administered agent. Bander et al. demonstrated acceptable toxicity and excellent targeting of known sites of metastases in 35 patients with androgen-independent PCa administered with [^177^Lu]Lu-J591. PSA response was seen in four patients, lasting from 3 to 8 months [25].

It took several years for the development and the first clinical experiences with the PSMA-targeting ligand PSMA-617. The first mCRPC patient was successfully treated with [^177^Lu]Lu-PSMA-617 at Heidelberg University Hospital by Kratochwil et al. [26].

### 2.2. Fundamentals of Radiotherapeutic Pharmaceuticals

Various compounds have been developed to target the PSMA receptor. One key difference lies in their size. Inhibitors of the glutamate–carboxypeptidase activity of the antigen, such as PSMA-617, are smaller compared to monoclonal antibodies, like J591, with sizes of 1 kDa and 150 kDa, respectively. This variance in molecular size influences pharmacokinetics significantly. PET scan biodistribution analysis revealed that only small molecules accumulate in salivary glands, independently from PSMA expression. Conversely, monoclonal antibodies exhibited longer circulation times. However, the quicker tumor uptake observed with PSMA-617 did not result in higher tumor-absorbed doses, as the therapeutic effect depends on the activity of the isotope, which decays over time [27]. Consequently, it is rational for scientific research in this domain to focus on achieving the optimal balance in radiopharmaceutical kinetics, which would profoundly impact the therapeutic window of these agents.

Several PSMA inhibitors have undergone clinical trials, including DOTA-PSMA-617 (used in VISION [28] and TheraP [29] trials) and DOTAGA-PSMA-I&T (utilized in SPLASH and ECLIPSE trials). Limited clinical trial data are available for another molecule, the nanobody PSMA-DOTA-JVZ-007 (JVZ-007) [30]. These molecules exhibit disparities in radiolabeling and subsequent biodistribution.

In vitro studies demonstrate good binding affinity and specificity of the tested PSMA-targeting tracers for the PSMA glycoprotein. The inability of PSMA nanobody JVZ-007 to be blocked by PSMA-617 and PSMA-I&T suggests that it binds to a different site, possibly explaining the different biodistribution. Compared to the other two radiopharmaceuticals, JVZ-007 exhibits lower biodistribution in human renal and salivary gland tissues but also lower in vitro tumor uptake, rendering it unsuitable for therapy [31].

PSMA-617 and PSMA-I&T present two different chelators, DOTA and DOTAGA, respectively, necessitating different radiolabeling reaction times and temperatures. To prepare the radiopharmaceuticals, PSMA-617 was incubated with [^177^Lu]LuCl_3_ for 8 min at 100 °C, while PSMA-I&T required a 30 min incubation at 95 °C [32]. Despite these differences in structure and labeling, in vivo biodistribution assays revealed comparable tumor uptake and induction of DNA double-strand breaks between [^177^Lu]Lu-PSMA-617 and [^177^Lu]Lu-PSMA-I&T. Following therapy with PSMA-617 and PSMA-I&T, blood tests indicated a decrease in hemoglobin levels, leukocytes, and platelets; though no clinical intervention was necessary. Both peptides exhibited the highest uptake in the lacrimal and parotid glands [33]. PSMA-617 demonstrated greater biodistribution throughout the body, while PSMA-I&T exhibited approximately 40 times higher renal uptake than PSMA-617, resulting in a less favorable tumor–kidney ratio.

For these reasons, PSMA-617 appears advantageous in terms of radiolabeling time and renal absorbed dose. Beyond the pharmacokinetics of the shuttle molecule, factors such as linear energy transfer (LET) and tissue range of the nuclide influence therapeutic effectiveness. A longer tissue range may be more useful in tumors with heterogeneous receptor expression, while a shorter range is safer in terms of absorbed dose to surrounding tissues, particularly in cases of extensive bone marrow infiltration. Nuclides with higher LET theoretically deliver localized doses in micrometastases, while those with lower LET rely on crossfire activity. Additionally, RLT is generally more effective in treating small metastases, as larger lesions may be necrotic, fibrous, poorly vascularized, and hypoxic, resulting in highly variable absorbed doses. Hence, the choice of the most suitable radiopharmaceutical may also depend on the stage of the disease being treated.

In this context, yttrium-90 (a high-energy β-emitter) and lutetium-177 (a medium-energy β-emitter) are currently the most widely used isotopes. A recent Monte Carlo simulation demonstrated that ^90^Y-labeled radiopharmaceuticals compensate for uptake heterogeneity within large tumors and effectively irradiate non-labeled targets. However, ^90^Y may not be ideal for eradicating micrometastases, as most of its energy is deposited outside the tumor, potentially reducing efficacy and increasing toxicity. Conversely, ^177^Lu has been reported to irradiate smaller spheres more effectively than ^90^Y [34].

Another isotope under investigation in the SECuRE clinical trial (NCT04868604) is copper-67 (^67^Cu), an electron emitter with a 62 h half-life. Its β^−^ emission has an energy of 121 keV, suitable for targeted radiotherapy of small tumors measuring 2–3 mm in size, while its γ-emission component can be utilized for SPECT-type imaging techniques. Radiolabeling ^67^Cu with PSMA-617 has shown excellent tissue distribution, high and sustained tumor uptake, and an outstanding tumor-to-tumor ratio, even in the early stages.

For the production of ^67^Cu, proton bombardment of enriched zinc-68 (^68^Zn) is considered the optimal approach. However, this method has disadvantages, such as the presence of radioisotope contaminants and natural copper contaminants in zinc targets, which can reduce specific activity [35].

In addition to β-emitting radiopharmaceuticals, other radionuclides have been explored for the treatment of mCRPC, including the α-emitter actinium-225 (^225^Ac) in the AcTION clinical trial [36]. Each radioisotope has its own advantages and disadvantages. Like ^177^Lu, ^225^Ac has a very long half-life (approximately 10 days) and can be easily complexed with the DOTA chelator, making it suitable for therapeutic purposes. During decay, ^225^Ac emits α particles characterized by lower tissue penetration and higher energy delivery compared to β^−^ particles emitted by lutetium. Alpha particles have the advantage of traveling only short distances, theoretically resulting in a promising target-therapy profile. Moreover, ^225^Ac requires very low activity for therapeutic use, limiting side effects due to toxicity.

Due to health physics regulations, only low levels of activity (<10 kBq) are permitted. When radiolabeling ^225^Ac, sensitive instrumentation is required to achieve the low impurity grade allowed by Good Manufacturing Practices (GMPs) (<1%), with equipment capable of detecting <100 Bq [37].

### 2.3. LuPSMA Trial

Between 2015 and 2016, the first prospective trial testing [^177^Lu]Lu-PSMA-617 was conducted at the Peter MacCallum Cancer Centre in Australia (LuPSMA trial by Hofman et al.). In this single-arm phase II trial, 30 PSMA PET-positive mCRPC patients, for whom conventional treatment failed, were enrolled. Patients received up to four doses of [^177^Lu]Lu-PSMA-617 (7.5 GBq) every 6 weeks. The primary endpoint (PSA decline greater than 50% from baseline value) was observed in 57% of patients. A total of 37% of patients experienced a significant improvement in global health score. The most common adverse events were grade 1 dry mouth, recorded in 87% of patients, transient nausea (50%), fatigue (50%), and grade 3 or 4 thrombocytopenia (13%). This trial finally raised commercial interest in PSMA RLT [38].

### 2.4. TheraP Trial

Soon after, a crucial phase II multi-center trial (TheraP trial) was conducted in Australia. The authors compared [^177^Lu]Lu-PSMA-617 therapy with a standard treatment in mCRPC patients, cabazitaxel. Between 2018 and 2019, 200 men were randomly assigned to two groups. Among them, 46% of the PSMA RLT patients completed all cycles, and 36% of the cabazitaxel group received all ten planned cycles. A PSA reduction of 50% or more from the baseline occurred more often in the [^177^Lu]Lu-PSMA-617 group (66% vs. 37%). Progression-free survival (PFS) at 12 months was 19% (95% CI 12–27) in the [^177^Lu]Lu-PSMA-617 group, while in the cabazitaxel group, it was 3% (1–9). Grade 3 or 4 adverse events occurred in 33% ([^177^Lu]Lu-PSMA-617 group) versus 53% (cabazitaxel group) of men. Grade 3–4 thrombocytopenia was more common in the [^177^Lu]-PSMA-617 group (11% vs. 0%). Grade 3–4 neutropenia was less common in the [^177^Lu]-PSMA-617 group (4% vs. 13%) [29].

It is worth mentioning that patients in this trial were carefully selected based on the combination of [^68^Ga]Ga-PSMA PET/CT and [^18^F]FDG PET/CT results to maximize the probability of observing benefit in the radioligand therapy group. Moreover, the population was not screened based on previous treatments. However, the promising efficacy and safety profile of [^177^Lu]Lu-PSMA-617 in previously treated patients has generated interest in exploring treatment combinations and the early use of [^177^Lu]Lu-PSMA-617 in the PCa natural history.

### 2.5. VISION Trial

In March 2021, during the ASCO World Congress 2021, results from an international, randomized, open-label phase III study, the VISION trial, were presented and laid the groundwork for a radical change in the future perspectives of PCa treatment and the history of Nuclear Medicine. The VISION trial evaluated the effects of [^177^Lu]Lu-PSMA-617 in PSMA-positive mCRPC patients [28]. PSMA positivity (upper threshold to the liver) was confirmed by a central review of [^68^Ga]Ga-PSMA-11 PET/CT images. Patients were randomized in a 2:1 ratio to receive [^177^Lu]Lu-PSMA-617 combined with standard of care therapy versus standard of care therapy alone. Standard treatment was determined by the investigator but excluded cytotoxic chemotherapy and [^223^Ra]-dichloride. Median radiological progression-free survival (rPFS) was 6 months for a hazard ratio (HR) of 0.67, and median OS was 13.7 months for an HR of 0.7306. Secondary endpoints included response rate according to Response Evaluation Criteria in Solid Tumors (RECIST) v1.1, disease control rate, and time to first symptomatic skeletal event.

During a median follow-up of 20.9 months, therapy exceeded a median increase in OS of 4.0 months, as well as rPFS of 5.3 months compared with standard treatment. The benefits of [^177^Lu]Lu-PSMA-617 therapy were consistent regardless of the standard treatment used, demographics, and disease characteristics. A 50% to 80% decrease in PSA was also much more common in the [^177^Lu]Lu-PSMA-617 group. These encouraging results led to the establishment of new phase III trials in patients who have not yet undergone chemotherapy at an earlier stage combining [^177^Lu]Lu-PSMA-617 with hormone therapy versus hormone therapy alone.

In conclusion, treatment with [^177^Lu]Lu-PSMA-617 together with standard therapy is a well-tolerated treatment that improves rPFS and OS compared to standard treatment alone in patients with PSMA-positive mCRPC in advanced disease, which could imply its possible implementation as standard treatment [28]. In March 2022, the FDA approved RLT with [^177^Lu]Lu-PSMA-617 for mCRPC based on the improvement in rPFS and OS observed in the VISION phase III trial.

### 2.6. First Steps in PSMA-Targeted α-Therapy: [^225^Ac]Ac-PSMA-617

Alongside the use of β-emitting isotopes, like ^177^Lu, the properties of α-emitting nuclides, like ^225^Ac, encouraged their therapeutic application. Rapid clearance of PSMA-617 limits renal translocation of daughter nuclides, which is the main problem for combination with antibodies, like J591, which were preliminarily studied with ^225^Ac-labeling [39]. The first study aimed at developing a treatment protocol for [^225^Ac]Ac-PSMA-617 was performed at Heidelberg University Hospital by Kratochwil et al. They extrapolated dosimetry empirically and observed that the most severe dose-limiting adverse event was xerostomia when treatment activity exceeded 100 kBq/kg per cycle [40]. The same group demonstrated remarkable anti-tumor activity in 40 heavily pre-treated mCRPC patients administered with 100 kBq/kg of [^225^Ac]Ac-PSMA-617 per three cycles performed every 2 months. PSA response was achieved in 63% of patients. Even after the approval of [^177^Lu]Lu-PSMA-617 to treat advanced-stage mCRPC, therapy using alpha emitters, such as [^225^Ac]Ac-PSMA-617, is also worth studying [39].

## 3. Trials Investigating Combination Therapies in Prostate Cancer

### 3.1. The Premises

Despite these promising results, both in the VISION [28] and TheraP [29] trials, primary resistance was reported in approximately 17–30% of patients treated with single-agent [^177^Lu]Lu-PSMA-617. One of the possible mechanisms that can explain this resistance is tumor heterogeneity. PCa is characterized by a long natural disease history during which different therapies are administered depending on tumor stage and aggressiveness. Clinicians can evaluate the aggressiveness of the tumor based on its invasion (localized, locally advanced, metastatic) and sensitivity to ADT. Usually, after the first hormone-sensitive phase, PCa cells become independent from androgens, necessitating the administration of a second-line therapy. PSMA expression can be considered another mechanism of resistance thanks to its catalytic roles of the poly-γ-glutamated folate that make folate available for cellular metabolism. The availability of a higher folate intracellular concentration is associated with increased tumor growth and higher invasiveness. PSMA expression also differs during PCa progression. It is higher in castration-resistant than in the hormone-sensitive phase and higher in metastatic than in localized PCa [41]. The encouraging results in terms of the clinical outcome and safety of [^177^Lu]Lu-PSMA-617 observed in mCRPC patients treated with [^177^Lu]Lu-PSMA after three lines of therapies set the ground for an application of PSMA-targeted therapy in the earlier stage of PCa, included as a neo-adjuvant therapy before radical prostatectomy. Another promising field of research is to combine RLT with other available PCa therapies, such as ARPIs, chemotherapy, immunotherapy, and targeted therapies (Table 1). The rationale is to strike the different pathways of tumor resistance to overcome tumor heterogeneity; at the same time, this approach can probably benefit from the effectiveness of each therapy and minimize the side effects.

### 3.2. The Trials

LuTectomy: [^177^Lu]Lu-PSMA-617 and Surgery [NCT04430192]

This open-label phase I/II non-randomized clinical trial studies the dosimetry, efficacy, and toxicity of PSMA-targeted therapy as a neoadjuvant therapy followed by radical treatment (radical prostatectomy and pelvic node dissection after 6 weeks) in high-risk localized or locoregional advanced PCa patients. This trial aims to determine the radiation absorbed dose in the prostate and involved lymph nodes and will evaluate the imaging response with PSMA-PET/CT. Biochemical response, pathological response, adverse effects of [^177^Lu]Lu-PSMA administration, surgical safety, and health-related Quality of Life (QoL) will be evaluated as secondary objectives. Preliminary results reported a median absorbed radiation dose of 19.6 Gy (Interquartile Range 11.3–48.4) in the prostate and 37.9 Gy (Interquartile Range 33.1–50.1) in lymph nodes. A partial response has been observed in 55% of patients, while 40% had stable disease, and 5% progressed during treatment; biochemical recurrence-free survival was 80% with a median follow-up of 13.8 months. The authors concluded that [^177^Lu]Lu-PSMA-617 followed by surgery is a safe and effective combination, characterized by high-targeted but variable radiation dose delivery.

ROADSTER: [^177^Lu]Lu-PSMA-617 and Brachytherapy [NCT05230251]

This phase II trial enrolls men with PCa experiencing biochemical recurrence after primary radiotherapy at least 2 years previously. The trial studies the safety and feasibility of one cycle of [^177^Lu]Lu-PSMA-617 followed by high-dose radiation brachytherapy to the entire prostate versus two brachytherapy treatments alone. The secondary outcome is the response rate to treatment, considered as a >50% decrease in PSA values.

POPSTAR II: [^177^Lu]Lu-PSMA-617 and Stereotactic Ablative Radiotherapy [NCT05560659]

This trial studies the effectiveness of the combination of [^177^Lu]Lu-PSMA with stereotactic ablative radiotherapy in oligometastatic patients. The primary outcome is PFS compared to stereotactic ablative radiotherapy alone. Oligometastatic PCa is considered a transition phase between localized and metastatic disease, where a curative intent is still considered possible. Inclusion criteria require the presence of 1–5 sites of nodal or bony metastases on [^68^Ga]Ga-PSMA or [^18^F]F-DCFPyL PET/CT. Interestingly, insensitivity of PSMA expression is required, with a score of 4 or 5 as defined by the E-PSMA criteria, and at least one site of disease with an SUVmax twice the SUVmax of the liver on PSMA PET must be detected (using [^68^Ga]Ga-PSMA-11 or [^18^F]F-DCFPYL tracers only).

PSMAaddition: [^177^Lu]Lu-PSMA-617 Plus Standard of Care [NCT04720157]

This trial studies the efficacy and safety of [^177^Lu]Lu-PSMA-617 in combination with ARPI and ADT (considered as standard of care) versus standard of care alone in mHNPC patients. The investigators expect to enroll approximately 1126 patients in this study. The primary outcome is the rPFS according to Prostate Cancer Clinical Trials Working Group 3 (PCWG3)-modified RECIST v1.1 criteria.

UpFront PSMA: [^177^Lu]Lu-PSMA-617 Plus Docetaxel [NCT04343885]

This is an open-label, randomized, stratified, 2-arm, multi-center phase II clinical trial that compares the effectiveness of [^177^Lu]Lu-PSMA-617 therapy followed by docetaxel chemotherapy versus docetaxel chemotherapy alone in patients with newly diagnosed high-volume HNPC. The primary outcome is the undetectable PSA rate at 12 months, defined as PSA ≤ 0.2 ng/mL.

LuCab: [^177^Lu]Lu-PSMA-617 Plus Cabazitaxel [NCT05340374]

This trial studies the safety of cabazitaxel in combination with [^177^Lu]Lu-PSMA-617 in mCRPC. Patients who have progressed on prior docetaxel and second-generation androgen inhibitor treatment are enrolled according to a prospective, single-center, single-arm, open-label phase I/II study design. The primary outcomes include determining the maximum tolerated dose (MTD), dose-limiting toxicities (DLTs), and recommended phase II dose of cabazitaxel in combination with [^177^Lu]Lu-PSMA-617.

[^177^Lu]Lu-PSMA Plus Pembrolizumab [NCT03805594; NCT05766371; NCT03658447]

The University of San Francisco (California) is conducting two trials on the combination of PSMA-targeted therapy and immunotherapy.

The first trial is an open-label, dose expansion phase I study aimed at determining the safety and effectiveness of a single dose of [^177^Lu]Lu-PSMA-617 followed by maintenance pembrolizumab in patients who have progressed on one or more ARPI and have at least three PSMA-avid lesions on [^68^Ga]Ga-PSMA-11 PET/CT [NCT03805594]. The protocol provides for the administration of a single dose of [^177^Lu]Lu-PSMA-617 (7.4 GBq) either 28 days before, concomitant with, or 21 days after the start of maintenance intravenous pembrolizumab (200 mg every 3 weeks). According to preliminary results, fourteen out of twenty-five patients (56%; 95% CI 35–76) had a PSA50 response, and four patients (16%; 95% CI 5–36) had a PSA90 response. The optimal timing for the administration of [^177^Lu]Lu-PSMA-617 was found to be 28 days before pembrolizumab [42]. The second trial is a phase II study in mCRPC patients progressing on one or more ARPI. The primary aim is to determine the 12-month rPFS rate by RECIST v1.1 and PCWG3 criteria [NCT05766371].

Another study, the PRINCE trial [43], examined the safety, tolerability, and efficacy of the combination of [^177^Lu]Lu-PSMA and pembrolizumab in patients with metastatic mCRPC. The inclusion criteria included a PSMA SUVmax ≥ 20 on at least one lesion and >10 at all metastatic sites. After a median follow-up of 16 months, the PSA50 response was demonstrated in 76% (28/37; 95% CI 59–88) of patients, and 70% (7/10) of patients showed an objective response according to RECIST v1.1 criteria. The median rPFS was 11.2 months (95% CI 5.1–14.1), and OS was 17.8 months (95% CI 13.4–not estimable). Toxicities were not clearly increased by the combination use of [^177^Lu]Lu-PSMA and pembrolizumab [NCT03658447].

LuPARP: [^177^Lu]Lu-PSMA-617 Plus Olaparib [NCT03874884]

This phase I, open-label, multicenter, dose escalation, and dose expansion study investigates the safety and tolerability of olaparib treatment in combination with [^177^Lu]Lu-PSMA in patients with mCRPC who have progressed on a novel ARPI regimen (abiraterone and/or enzalutamide and/or apalutamide). Patients with prior treatment with platinum agents, PARP inhibitors, or [^177^Lu]Lu-PSMA are excluded, while prior administration of docetaxel in the chemotherapy-naïve setting or castrate setting is allowed. The primary outcomes are the DLTs and MTD during the escalation phase and the recommended dose for the second phase.

EVOLUTION: [^177^Lu]Lu-PSMA-617 Ipilimumab and Nivolumab [NCT05150236]

This is an open-label, randomized, stratified multicenter phase II clinical trial that investigates the activity and safety of [^177^Lu]Lu-PSMA therapy versus [^177^Lu]Lu-PSMA in combination with ipilimumab and nivolumab in patients with mCRPC. Ipilimumab and nivolumab are two different kinds of immunotherapy drugs. Ipilimumab is a fully humanized IgG1 monoclonal antibody that blocks cytotoxic T lymphocyte antigen-4 (CTLA-4) and is indicated in metastatic or unresectable melanoma. Nivolumab is a fully human IgG4 antibody targeting the immune checkpoint programmed death receptor-1 (PD-1) used to treat melanoma, non-small-cell lung cancer, renal cell cancer, head and neck cancer, and Hodgkin lymphoma.

In this trial, patients are randomized in a 2:1 ratio and stratified by prior exposure to docetaxel. The primary outcome is PSA-PFS at 1 year, defined as a rise in PSA by ≥25% and ≥2 ng/mL above the nadir (lowest PSA point) confirmed by a repeated PSA at least 3 weeks later.

CaboLu: [^177^Lu]Lu-PSMA-617 Plus Cabozantinib [NCT05613894]

This is an open-label phase I dose escalation study of cabozantinib in combination with [^177^Lu]Lu-PSMA-617 in subjects with mCRPC. Cabozantinib is a tyrosine kinase inhibitor indicated in various kinds of tumors, such as kidney, thyroid, and liver cancer. The trial is composed of two phases. During the first phase, dose escalation will define the rate of DLTs; in the second phase, the MTD and/or recommended dose and schedule will be assessed. The primary outcomes are the rate of DLTs during the DLT evaluation period and the proportion of patients without progression as defined by PCWG3-modified RECIST v1.1 at 24 weeks. The MTD, recommended dose, and schedule for the subsequent expansion stage of daily oral administration of cabozantinib will be estimated based on the occurrence of DLTs.

LuPIN: [^177^Lu]Lu-PSMA-617 and Idronoxil

This trial studied the safety and activity of [^177^Lu]Lu-PSMA-617 in combination with idronoxil (NOX66) in patients with end-stage mCRPC. Idronoxil is a synthetic flavonoid derivative that promotes apoptosis by binding to the transmembrane enzyme ENOX2 expressed on cancer cells and has shown radiosensitizing properties. Thirty-two mCRPC patients who progressed on taxane-based chemotherapy and abiraterone and/or enzalutamide were enrolled. Patients received up to six cycles of [^177^Lu]Lu-PSMA-617 combined with escalating doses of 400 mg (cohort 1) or 800 mg (cohort 2) of NOX66. A PSA50 response was reported in 62.5% of the patients (95% CI 45–77), while 91% had any PSA response (95% CI 76–97). The median PSA-PFS was 6.1 months (95% CI 2.8–9.2), and the median OS was 17.1 months (95% CI 6.5–27.1).

Fractionated [^177^Lu]Lu-J591 Antibody Plus Docetaxel [NCT00916123]

This is a phase I study in mCRPC patients who progressed despite adequate medical or surgical castration therapy. Patients who had previously received docetaxel must not be in progression, and all docetaxel-related toxicities must have resolved to <grade 1. No PSMA imaging criteria were required for enrollment. The fifteen enrolled patients received two cycles of docetaxel followed by two fractionated doses of [^177^Lu]Lu-J591 concurrent with the third cycle of docetaxel, followed by ongoing docetaxel every 3 weeks. A PSA50 response and a partial response according to RECIST 1.1 criteria were reported in 73% and 60% of the patients, respectively. No dose-limiting toxicity was seen at any dose level, suggesting that a single fractionated cycle of [^177^Lu]Lu-J591 combined with docetaxel/prednisone is feasible in patients with mCRPC.

[^225^Ac]Ac-J591 Antibody Plus Pembrolizumab and Androgen Receptor Pathway Inhibitor [NCT04946370]

This is a phase I/II study investigating the combination of [^225^Ac]Ac-J591 with pembrolizumab in mCRPC. The study aims to determine the optimal dose of [^225^Ac]Ac-J591 in combination with pembrolizumab (phase I) and then assess the effectiveness of the combination of [^225^Ac]Ac-J591, pembrolizumab, and ARPI compared to pembrolizumab and ARPI alone (phase II). Patients will be considered responders if at least one of the composite criteria is satisfied, including a reduction in PSA values, a decrease in circulating tumor cell count, and imaging changes observed on a CT and bone scan. A [^68^Ga]Ga-PSMA-11 PET/CT scan will be performed prior to therapy and at 12 weeks.

## 4. Future Perspectives

As described above, several factors contribute to determining the effectiveness of RLT: intrinsic tumor characteristics, patient selection criteria, tumoral cellular targets, radioisotopes, and related physical characteristics used. We have seen how the combination of multiple treatments can play a crucial role in the management of patients with PCa, a neoplasm generally characterized by a long clinical course, multiple lines of therapy, and intrinsic tumor heterogeneity. Nevertheless, numerous studies are ongoing to obtain additional tools available to clinicians.

PSMA molecules have been labeled with other α-particle emitters, like astatine-211 [44,45,46], terbium-149 [47,48,49], and lead-203 [50,51,52,53]. Among these, an interesting ongoing phase I clinical trial on mCRPC patients is evaluating the safety, tolerability, pharmacokinetics, and anti-tumor activity of a ^227^Th-labeled PSMA-specific monoclonal antibody alone or in combination with darolutamide.

Advantages in terms of efficacy can also be introduced by optimizing the pharmacokinetic profile of PSMA ligands. Two prospective phase I clinical trials on mCRPC patients reported longer blood circulation of [^177^Lu]Lu-PSMA when combined with albumin molecules ([^177^Lu]Lu-PSMA-ALB-56) [54], leading to increased tumor uptake. No severe adverse events were observed using albumin-binding PSMA RLT, but a lower incidence of hematological toxicities has been reported in a preclinical trial when these radiopharmaceuticals were conjugated with ibuprofen [55,56]. This can be explained by more efficient clearance from the blood pool to keep background activity in healthy tissues as low as possible.

In addition to PSMA, other studies are exploring the potentiality of new promising targets overexpressed on PCa cells, such as the gastrin-releasing peptide receptor (GRPR). A multicenter clinical trial is evaluating the feasibility of ^177^LuNeoBOMB1 in patients with advanced solid tumors known to overexpress GRPR (using bombesin analog peptides as a vector) [NCT03872778]. Promising results also come from another study on 35 mCRPC patients treated with [^177^Lu]Lu-RM2, a GRPR antagonist [57].

Another challenge in PCa is the management of neuroendocrine-differentiated adenocarcinomas characterized by lower expressions of androgen receptors and PSMA, especially after treatment with AR inhibitors, which does not allow advantageous use of PSMA-targeted therapies. In these patients, the use of radiopharmaceuticals targeting the somatostatin receptor (SSTR) [58,59,60] or delta-like ligand 3 [61,62] seems to be promising.

## 5. Conclusions

PSMA-targeted therapy provides added value to the therapeutic options for PCa patients. Alongside the recognized indication in the metastatic castration-resistant phase after progression to taxane-based treatments, several clinical trials are investigating the efficacy of RLT in other clinical settings and in combination with other PCa treatments. In our opinion, the future perspective of PSMA-targeted therapy relies more on the intensity of tumor cell PSMA expression than on the clinical setting. According to these criteria, patient selection would be more stringent in the earlier stages of the disease and less restrictive in the later stages, when RLT can be combined with other PCa treatments to overcome tumor resistance patterns and heterogeneity. In this scenario, multidisciplinary management of these patients will become even more crucial.

## Figures and Tables

**Table 1 cancers-16-01643-t001:** Trials investigating combination therapies in prostate cancer.

Study Title	Disease Stage	Treatment Combined with RLT	Study Phase	Primary Outcome Measures
Dosimetry, Safety and Potential Benefit of [^177^Lu]Lu-PSMA-617 Prior to Prostatectomy (LuTectomy)	High-risk localized or locoregional advanced PCa	Surgery	Phase I/II Active	- Radiation absorbed the dose in the prostate and involved lymph nodes following one or two administrations of [^177^Lu]Lu-PSMA
Radioligand for locAl raDiorecurrent prostate cancer (ROADSTER)	Locally recurrent PCa	Brachytherapy	Phase II Active	- Safety and efficacy
[^177^Lu]Lu-PSMA for Oligometastatic Prostate Cancer Treated with STereotactic Ablative Radiotherapy (POPSTAR II)	Oligometastatic PCa	SABR	Phase II Active	- Evaluate the (bPFS) of SABR alone and SABR + [^177^Lu]Lu-PSMA
International Prospective Open-label, Randomized, Phase III Study Comparing [^177^Lu]Lu-PSMA-617 in Combination with SoC, Versus SoC Alone, in Adult Male Patients with mHSPC (PSMAddition)	mHSPC	SoC	Phase III Active Not Recruiting	- rPFS is defined as the time of radiographic progression by PCWG3-modified RECIST V1.1
Randomized phase 2 study of sequential [^177^Lu]Lu-PSMA-617 and docetaxel vs docetaxel in metastatic hormone-naïve prostate cancer (clinical trial protocol) (UpFrontPSMA)	mHNPC	Docetaxel	Phase II Active	- Undetectable PSA rate at 12 months after commencement of protocol therapy (defined as PSA ≤ 0.2 ng/mL)
Cabazitaxel in Combination with [^177^Lu]Lu-PSMA-617 in Metastatic Castration-resistant Prostate Cancer (LuCAB)	mCRPC	Cabazitaxel	Phase I/II Active	- MTD, DLT, and recommended phase II dose of cabazitaxel in combination with [^177^Lu]Lu-PSMA
[^177^Lu]Lu-PSMA-617 and Pembrolizumab in Treating Patients with Metastatic Castration-Resistant Prostate Cancer [42]	mCRPC	Pembrolizumab	Phase Ib Active	- Recommended phase II dose - ORR by Response Evaluation Criteria in Solid Tumors by RECIST 1.1 criteria
Pembrolizumab Plus [^177^Lu]Lu-PSMA-617 in Patients with Castration Resistant Prostate Cancer	mCRPC	Pembrolizumab	Phase II Active	- Recommended phase II dose and schedule - ORR by RECIST 1.1 criteria
PSMA-lutetium Radionuclide Therapy and Immunotherapy in Prostate Cancer [43] (PRINCE)	mCRPC	Immunotherapy	Phase Ib/II	- PSA response - Incidence of treatment-emergent AEs - Tolerability
[^177^Lu]Lu-PSMA-617 Therapy and Olaparib in Patients with Metastatic Castration Resistant Prostate Cancer (LuPARP)	mCRPC	Olaparib	Phase I Active Recruiting	- DLTs and MTD (phase I) - Recommended dose (phase II)
[^177^Lu]Lu-PSMA Therapy Versus [^177^Lu]Lu-PSMA in Combination with Ipilimumab and Nivolumab for Men with mCRPC (EVOLUTION)	mCRPC	Ipilimumab and Nivolumab	Phase II Active Recruiting	- PSA-PFS at 1 year (PCWG3) is defined as a rise in PSA by ≥25% and ≥2 ng/mL above the nadir (lowest PSA point), confirmed by a repeated PSA at least 3 weeks later
Dose-Escalation Study of Cabozantinib in Combination with [^177^Lu]Lu-PSMA-617 in Patients with Metastatic Castration-Resistant Prostate Cancer (CaboLu)	mCRPC	Cabozantinib	Phase I Active	- The rate of DLTs during the DLT evaluation period - The proportion of patients without progression as defined by PCWG3-modified RECIST 1.1 at 24 weeks
Phase I/II Trial of the Combination of [^177^Lu]Lu-PSMA-617 and Idronoxil (NOX66) in Men with End-stage Metastatic Castration-resistant Prostate Cancer (LuPIN)	mCRPC	Idronoxil	Phase I/II Concluded	- AEs, pain inventory scores, PSA response, PFS
Docetaxel/Prednisone Plus Fractionated [^177^Lu]Lu-J591 Antibody for Metastatic, Castrate-resistant Prostate Cancer	mCRPC	Docetaxel and Prednisone	Phase I Concluded	- MTD of fractionated [^177^Lu]Lu-DOTA-J591 administered concurrently with docetaxel three times weekly
Phase I/II Trial of Pembrolizumab and Androgen-receptor Pathway Inhibitor wth or without [^225^Ac]Ac-J591 for Progressive Metastatic Castration Resistant Prostate Cancer	mCRPC	Pembrolizumab and ARPi	Phase I/II Active	- Proportion of patients with DLT following treatment with pembrolizumab and [^225^Ac]Ac-J591 - Determination of the optimal dose of [^225^Ac]Ac-J591 for phase II - Change in the composite response rate of pembrolizumab and ARPI with or without [^225^Ac]Ac-J591

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
