# Peer review of "Prostate-Specific Membrane Antigen-Targeted Therapy in Prostate Cancer: History, Combination Therapies, Trials, and Future Perspective"

_cancers, 2024, doi:10.3390/cancers16091643_

Round 1

Reviewer 1 Report (Previous Reviewer 1)

Comments and Suggestions for Authors

I do not have any scientific concern and the article can be accepted. However, you might want to get it read by a language editor before making a final decision. For example, I noticed a couple of errors:

Abstract : ‘i’ instead of ‘I’
Line 262: ‘usinh’ instead of ‘using’.

Comments on the Quality of English Language

The article needs to be extensively edited for language and perhaps.

Author Response

The authors thank the reviewer for the valuable suggestions.

Reviewer 1

Abstract: ‘i’ instead of ‘I’: we believe the reviewer is referring to the “i” at the beginning of the paragraph, which was written in lowercase because it follows a colon. Nevertheless, the text has been modified.

Line 262: ‘usinh’ instead of ‘using’: this typing error has been amended.

Reviewer 2 Report (Previous Reviewer 2)

Comments and Suggestions for Authors

The revised version of the paper has suitably addressed the issues raised, and the rewriting of some paragraphs has improved the paper. A few issues have been noted, but these are all very minor.

1. Line 154: “Nuclides with higher LET theoretically deliver effective doses in micrometastases”. The term “effective dose” has an established meaning related to whole-body radiation dose, which does not seem to be the meaning here. It is therefore suggested to use a different wording, for instance “concentrated doses” or “localized doses”.

2. Line 345: It is suggested to add both of the relevant clinical trial numbers: NCT05766371 and the clinical trial number for the PRINCE study (this clinical trial number does not seem to appear anywhere in the text?).

3. Table 1: It is suggested to include the clinical trial numbers in the first column.

TYPOGRAPHICAL

Section headline typography: Headlines 3 and 4 are in all-uppercase, while headlines 1, 2, and 5 have only the initial letter in uppercase. Is this difference on purpose?

Line 119: It is suggested to write “did not” instead of “didn’t”. While abbreviations like “didn’t”, “don’t”, “he’s”, etc. are very common in spoken English, they are normally avoided in written (formal) English.

Line 121: it’s -> it is

Line 168: KeV -> keV (it would be logical to have uppercase K for kilo- in line with M for mega- and G for giga-, but likely for historical causes, this is not the standard; a kilogram is written kg, not Kg)

Line 231: “223” should be uppercase in 223Ra.

Line 262: usinh -> using

Table 1: For the EVOLUTION study, there is an unwanted line shift in the rightmost column.

Line 413: There is an unwanted space in “Lu- J591”

Author Response

The authors thank the reviewer for the valuable suggestions.

  1. Line 154: “Nuclides with higher LET theoretically deliver effective doses in micrometastases”: the term “effective dose” has been changed with “localized dose”.
  2. Line 345: It is suggested to add both of the relevant clinical trial numbers: the clinical trial numbers have been included.
  3. Table 1: It is suggested to include the clinical trial numbers in the first column: for improved readability of the table, we chose not to include the trial numbers in the first column.

TYPOGRAPHICAL

Section headline typography: amended.

Line 119: It is suggested to write “did not” instead of “didn’t”: amended.

Line 121: it’s -> it is: amended.

Line 168: KeV -> keV: amended.

Line 231: “223” should be uppercase in 223Ra: amended.

Line 262: usinh -> using: amended.

Table 1: For the EVOLUTION study, there is an unwanted line shift in the rightmost column: amended.

Line 413: There is an unwanted space in “Lu- J591”: amended.

Reviewer 3 Report (Previous Reviewer 3)

Comments and Suggestions for Authors

This is a re-review of revision of paper.  Appreciate the authors attention to the referee's comments.  In general, these have been addressed to improve the paper.  I do not see any outstanding issues otherwise.

Author Response

The authors thank the reviewer for the comments.

This manuscript is a resubmission of an earlier submission. The following is a list of the peer review reports and author responses from that submission.

Round 1

Reviewer 1 Report

Comments and Suggestions for Authors

The article needs to be extensively edited for language and perhaps re-written purely from the language point of view. I have no concern about the scientific nature of the article. From the language point of view, I am including some examples, but this is applicable to the entire article:

1.       The title needs to be changed from “PSMA-target therapy in prostate cancer” to “PSMA-targeted therapy in prostate cancer:..”

2.       Line 13: Radioligand Therapy play” should be “Radioligand Therapy plays”

3.       Line 24: both in staging than in recurrence , should his be “both in staging and recurrence?”

4.       Line 27: ..and put the basis for….

5.       Lines 41 and 42: ..come up against a recurrence….

6.       Line 56: In 2004 docetaxel demonstrate to prolong survival

Comments on the Quality of English Language

The article needs to be extensively edited for language and perhaps re-written purely from the language point of view. I have no concern about the scientific nature of the article. From the language point of view, I am including some examples, but this is applicable to the entire article:

1.       The title needs to be changed from “PSMA-target therapy in prostate cancer” to “PSMA-targeted therapy in prostate cancer:..”

2.       Line 13: Radioligand Therapy play” should be “Radioligand Therapy plays”

3.       Line 24: both in staging than in recurrence , should his be “both in staging and recurrence?”

4.       Line 27: ..and put the basis for….

5.       Lines 41 and 42: ..come up against a recurrence….

6.       Line 56: In 2004 docetaxel demonstrate to prolong survival

Author Response

The authors thank the reviewer for the valuable comments, which they hope have been addressed as comprehensively as possible.

Reviewer 2 Report

Comments and Suggestions for Authors

The paper "PSMA-target therapy in prostate cancer: history, combination therapies tries and future perspective” by Dr. Mattana et al. is a combination of two reviews that supplement each other: section 2 reviews the history of how 177Lu-labeled PSMA became FDA-approved as prostate cancer (PCa) theapy, while section 3 reviews the newer and ongoing study of how radiopharmaceuticals can be combined with other therapies for PCa therapy. The combined review works well but needs some polishing in its impression. Below is given a list of specific issues to polish or consider, but all are minor.

MINOR

Notation for radiotracers: A widely accepted consensus paper with guidelines for radiochemistry nomenclature was published in 2017 and summarized in a widely published open letter that can be found for instance here: https://doi.org/10.1186/s41181-018-0047-y . The European Association of Nuclear Medicine (EANM) has abbreviated it to a short guideline on their web page: eanm.org/publications/guidelines/nomenclature/ 
According to the guideline, radiopharmaceuticals should be written with the full molecule (including the atom used for labelling), with the specific isotope in square parentheses right to the left of this. Thus, while [18F]FDG is correct (FDG is the full molecule, including the fluorine atom), 177Lu-labeled PSMA-617 should be written [177Lu]Lu-PSMA-617 (the full molecule is Lu-PSMA-617, while PSMA itself does not include lutetium). Similarly for the other ligands mentioned and for [223Ra]Ra-dichloride (or [223Ra]radiumdichloride).

Choice of term for the therapy: The review uses the term "radioligand therapy (RLT)". For therapy involving radionuclides, the term "radiopharmaceutical therapy (RPT)" is increasingly being used. Citing from ICRP Publication 140, Radiological Protection in Therapy with Radiopharmaceuticals: "Therapy with radiopharmaceuticals is referred to by many synonymous terms, including ‘targeted radionuclide therapy’, ‘unsealed source therapy’, ‘systemic radiation therapy’, and ‘molecular radiotherapy’. In this publication, the generic term ‘radiopharmaceutical therapy’ is used for consistency with other ICRP and ICRU publications." (Footnote on page 13 of IRCP 140). It is suggested (but not requested) that the authors consider using the term "radiopharmaceutical therapy (RPT)" rather than "radioligand therapy (RLT)".

Upper and lower case: The use of upper and lower case is not quite consistent. Unless the context calls for uppercase of any word (e.g., first word of a sentence), elements, like lutetium, actinium, astatine, etc. are by standard written with lower case initial letter (but of course Lu, Ac, At, etc. for the chemical abbreviations). Likewise, antibodies etc. like pembrolizumab are usually written with lower case, reserving uppercase for trade names. Please check the capitalization of words (especially in section 3, including Table 1).

Abbreviations: PSA (prostate-specific antigen) should be defined at first use in line 101. HR (hazard ratio?) should be defined at first use in line 240. In Table 1, “PC” is used several times without definition, apparently it should be PCa?

Line 141 (see also lines 144-145): 177LuCl3 -> [177Lu]LuCl3 (with both superscript and subscript)

Line 169: outside the tumor -> outside such small tumors [to make the point more clear]

Line 271: The second half of the sentence is difficult to read, and “TAT” (targeted alpha therapy) is not defined and can be mistaken as being part of the tracer name. Suggested rephrasing: “... mCRPC, therapy using alpha emitters such as [225Ac]Ac-PSMA-617 is also worth studying [39].”

Table 1, row 11: Is the study name EVOLUTION or ANZUP2001? (The study is referred to as ANZUP2001 in line 383).

Table 1, row 13: The study registration number is missing.

Line 353: The study registration number is missing from the sub-headline.

Line 367: A sub-headline should be added here to introduce the PRINCE study.

In many cases in the study descriptions in section 3.2, the text describes what the study “will” do, corresponding to future actions. However, as most of these studies are active, present tense will at most places be a better description, e.g., “the tries studies” instead of “the trial will study”.

Lines 448-449: In the list of alpha-particle emitters, please list not just the element, but the specific radionuclide (each of astatine, terbium, and lead exist as several isotopes, and not all of these are radioactive).

Abbreviation list: Please sort the list alphabetically.

TYPOGRAPHICAL

Table 1 is difficult to read because of the indentation after first line in many cells.

Much of the text in Table 1 and in the following descriptions in the main text is in a slightly different color than black (more obvious when printing as gray). Please make all text standard black.

In section 3, the text is inconsistently formatted, with varying font size and line size. Please use consistent formatting.

Line 164 and other places: Should “β-“ be understood as “beta-minus” (not beta-plus)? If yes, the minus should be in superscript. If it is a hyphen, not a minus sign, it should be one word: β-emitter (“beta-emitter”)

Line 408: In this sub-headline, the trial number is round parentheses ( ), while in all the other sub-headlines, square brackets [ ] are used.

Comments on the Quality of English Language

GRAMMAR

In a number of places, the grammar needs correction:

Line 13: play -> plays [Radioligand therapy is singular]

Lines 21 and 74: radiopharmaceutical -> radiopharmaceuticals [not just one radiopharmaceutical, and if it was, it should be “a ... radiopharmaceutical”]

Line 48: patient’s (singular genitive) -> patients’ (plural genitive)

Line 56: docetaxel demonstrate -> docetaxel was demonstrated to

Line 77: in the most histotype of PCa cells beside to -> in most histotypes of PCa cells compared to [“the most histotype” makes “histotype” an adjective rather than a noun]

Line 78: become -> becomes [PSMA is singular]

Line 92: where is the headquarter of the PCF -> where the headquarter of PCF is located [or “the PCF” as originally – in that case, add also the definite article in line 88, “the Prostate Cancer Foundation (PCF)” in line 88.]

Line 93: was -> were [antibodies are plural]

Line 95: surface inside the cell -> inner cell surface

Line 104: 177Lutetium -> lutetium-177 (177Lu) [including the parenthesis]

Line 111: Radiopharmaceutical’s fundamentals -> Radiopharmaceutical fundamentals [no ’s]

Line 113: The first difference stay in size -> The first difference is size

Line 113: inhibitor -> inhibitors [Question: is this meant as a general statement about all inhibitors and antigens? If it is meant more specifically about those used, write in the definite form: “the inhibitors ... are smaller than ... the antigens”.]

Line 116: with PET scan -> with PET scanning [also possible: with a PET scan]

Line 119: didn’t -> did not [this form of abbreviation is very common in spoken English, but usually avoided in written English]

Line 119: a ... doses -> a ... dose [or: a ... doses -> ... doses]

Line 163: can relies -> can rely

In many places: emittent -> emitter

Line 232: put the basis -> lay the foundation [or lay the basis]

Line 317: trial study -> trial studies [trial is singular]

Line 343: compare -> compares

Author Response

(The authors gave the same response as above.)

Reviewer 3 Report

Comments and Suggestions for Authors

Mattana et al present a review article regarding PSMA targeting in prostate cancer.  PSMA targeted therapy is the latest class of agent approved in prostate cancer and has provided a new treatment option for patients.  It is already playing an important role and many studies are underway to define its optimal use.  This review discussed the background of PSMA targeting, some historical agents that have been used, and reviewed clinical data and ongoing clinical trials.  Overall appears to be comprehensive.  English editing and flow of language can be improved.  The perspective from authors can be expanded.

Specific comments:

1. Page 2 line 55-56: FDA approval for olaparib is more broad than BRCA1/2 (although European guidelines may restrict to that population).  Need to also mention other PARP inhbitors now improved more broadly, both single agent and combination.

2. Page 2 line 78: unclear meaning of "curative intent" do you mean target for therapy?

3. In background on PSMA targeting, should also discuss other antibody drug conjugates with J591 -- not all PSMA targeting molecules have proven successful in the clinic.  For example also discuss MLN2704, PSMA ADC (w/MMAE) where there is phase 1/2 data.

4. In general, when discussing each of the clinical trials -- most importantly those with phase 2/3 clinical data such as VISION, include more much extensive detail regarding inclusion/exclusion, allowed therapies, and criteria for PSMA positivity (if included).

5. PSMAfore should be discussed.  This data is available in abstract form (presented at ESMO 2023).

6. In Table 1: PSMA positivity criteria may be helpful to be included

7. For both the Table and following discussion of pending studies, it would be helpful to reader for thematic grouping/organization of these studies.  For example, classes of combination agents or disease states.

8. The discussion/perspective could be significantly expanded.  Where do authors see as best and most promising clinical situation for Lu-PSMA use? 

Comments on the Quality of English Language

many sentences are coarsely constructed and editing would be recommended.

Author Response

(The authors gave the same response as above.)

Reviewer 4 Report

Comments and Suggestions for Authors

F. Mattana, L. Muraglia, A. Barone, M. Colandrea, Y. S. Diffalah, S. Provera, A. S. Cascio, E. O. Salè and F. Ceci, PSMA-target therapy in prostate cancer: history, combination therapies trials and future perspective

To write a helpful review is a heavy task. The authors should inform both the beginners and experts of the concrete field. Here, we find a summary about the history of PSMA-target therapies, but something is missing. For example: I encourage the authors inserting a figure showing the main characteristics of PSMA-target therapy. Problems appear mostly in the third point of the manuscript (3. Combination therapies). After a short introduction (3.1. The premises) the authors summarized fifteen trials in Table 1. The appearance of this Table is not advantageous. I suggest considering the recently published works (reviews, even in 2023) and change not only the form of the Table but the number of columns and the descriptions of the clinical trials. The authors described the trials heterogeneously in 3.2. Duplicated information is in Table 1 and point 3.2. The reader can find two lists of abbreviations (under Table 1 and in the last section of the manuscript). I recommend completing this list. Moreover, I suggest adding an extra paragraph in the last part of point 3.2. (before 4. Future perspectives) to give comments about the trials.

Author Response

(The authors gave the same response as above.)
